# Crop water stress maps for an entire growing season from visible and thermal UAV imagery

Helene Hoffmann[1], Rasmus Jensen[1], Anton Thomsen[2], Hector Nieto[3], Jesper Rasmussen[4] and Thomas Friborg[1]

[1]Department of Geosciences and Natural Resource Management, University of Copenhagen, Øster Voldgade 10, 1350 Cph. K, Denmark

[2]Department of Agroecology, Aarhus University, Nordre Ringgade 1, 8000 Aarhus C, Denmark

[3]Instituto de Agricultura Sostenible (IAS) Consejo Superior de Investigaciones Científicas (CSIC), Campus Alameda del Obispo, Av. Menéndez Pidal s/n, 14004 Córdoba, Spain

[4]Department of Plant and Environmental Sciences, University of Copenhagen, Højbakkegaard Allé 9, 2630 Taastrup, Denmark

*Correspondence to*: Helene Hoffmann (helene.hoffmann@ign.ku.dk)

**Abstract.** This study investigates whether a Water Deficit Index (WDI) based on imagery from Unmanned Aerial Vehicles (UAVs) can provide accurate crop water stress maps at different growth stages of barley and in differing weather situations. Data from both the early and the late growing season are included to investigate whether the WDI index has the unique potential to be applicable both when the land surface is partly composed of bare soil and when crops on the land surface are senescing. The WDI index differs from the more commonly applied Crop Water Stress Index (CWSI) in that it uses both a spectral vegetation index (VI), to determine the degree of surface greenness, and the composite land surface temperature (LST) (not solely canopy temperature).

Lightweight thermal and RGB (Red-Green-Blue) cameras were mounted on a UAV on three occasions during the growing season, 2014, and provided composite LST and color images, respectively. From the LST, maps of surface-air temperature differences were computed. From the color images, the Normalized Green-Red Difference Index (NGRDI), constituting the indicator of surface greenness, was computed. Advantages of the WDI as an irrigation map, as compared with simpler maps of the surface-air temperature difference, are discussed, and the suitability of the NGRDI index is assessed. Final WDI maps had a spatial resolution of 0.25 m.

It was found that the UAV-based WDI index is in agreement with measured stress values from an Eddy Covariance system. Further, the WDI index is especially valuable in the late growing season because at this stage the remote sensing data represent crop water availability to a greater extent than they do in the early growing season, and because the WDI index accounts for areas of ripe crops that no longer have the same need of irrigation. WDI maps can potentially serve as water stress maps, showing the farmer where irrigation is needed to ensure healthy growing plants, during entire growing seasons.

# 1 Introduction

Recent developments in Unmanned Aerial Vehicles (UAV's) have extended the practice of remote sensing in precision agriculture research (Berni et al., 2009b; Gonzalez-Dugo et al., 2012; Hoffmann et al., 2016; Turner et al., 2011; Vergara-Díaz et al., 2016; Zarco-Tejada et al., 2009, 2013b). UAV platforms now enable the collection of remotely sensed temperatures with higher spatial and temporal resolution than those collected by satellites and manned aircraft. Temperatures from canopies are closely related to air and soil water content, actual transpiration and crop water stress (Idso et al., 1986; Jackson et al., 1987). If a crop has insufficient water supply, stomata will close in order to limit water loss through transpiration. This leads to energy being stored and thus higher canopy temperatures, relative to those seen in crops with ample water supplies (Guilioni et al., 2008). Canopy temperature is therefore positively correlated with crop water stress and negatively correlated with soil moisture and transpiration. Quantifying crop water stress allows farmers to assess the need for irrigation. Idso et al., 1981 and Jackson et al., 1981 developed the well documented and commonly used Crop Water Stress Index (CWSI) (Barbosa da Silva and Ramana Rao, 2005; Feldhake et al., 1997; Sepaskhah and Kashefipour, 1994; Xiang and Tian, 2011). They related canopy temperatures to evaporation and presented the stress index as $1 - \frac{\lambda E_{act}}{\lambda E_{pot}}$, where E is the evapotranspiration rate (mm) and λ is the latent heat of vaporization of water (MJ m$^{-2}$). λE is the latent heat flux density (W m$^{-2}$).

Practical applications of the CWSI have been hampered by difficulties in measuring canopy temperature alone in fields with partial vegetation cover. Remotely sensed land surface temperatures (LSTs) from partially vegetated fields such as arable crops in early growth stages are composites of soil and canopy temperatures, and separating the two source temperatures is problematic (Luquet et al., 2004). High resolution thermal images from crops with large canopies allow pure canopy pixels, and hence canopy temperatures, to be isolated. Consequently, most recent research in UAV imagery and crop water stress have focused on grapevines, olive trees and other crops with relatively large canopies (Berni et al., 2009a; Gago et al., 2013, 2015; Gonzalez-Dugo et al., 2013; Zarco-Tejada et al., 2013a). Crops with open canopies in early growth stages, such as barley and other cereals, result in imagery with composite temperatures, even on spatially high resolution images. In order to eliminate the process of separating soil and canopy temperatures, Moran et al. (1994) developed the Water Deficit Index (WDI). In addition to using the composite LST (composite soil and canopy temperature), this also uses a spectral vegetation index (VI) to estimates the surface greenness, i.e. the fraction of canopy cover.

In this study, a UAV-based WDI index was applied to barley fields three times during the spring and summer of 2014 in order to assess whether it can detect intra-field variations of crop water status at different crop growth stages. Both the early and the late growing seasons were investigated to assess whether the WDI index possesses the unique potential to give accurate results both when the surface is partially composed by bare soil and when the crops are senescing. The latter required the WDI to be based on a VI that can determine the greenness of crops, i.e. whether they are ripe or not ripe. We classified yellow barley as ripe or prematurely ripe and green barley as not ripe. Thus in this study, the VI served two purposes: detection of the canopy cover in the early growing season, as originally intended, and detection of the

developmental stage of crops in the late growing season. The Normalized Green-Red Difference Index (NGRDI) was used as the VI, and its suitability to be incorporated in the WDI index will be discussed below. Further, we extended the original WDI index with an extra set of thermal observations in order to accommodate any offset between surface and air temperature measurements caused by, for example, atmospheric effects (Anderson et al., 1997). The composite LST and the VI's (on

which the WDI was based) were collected in both overcast and moderately sunny weather conditions.

The objective was thus to investigate whether a WDI index based on UAV imagery can provide accurate crop water stress maps during different growth stages of barley and in different weather situations.

## 2 Materials and Methods

### 2.1 Site

This study was carried out in two adjacent spring barley fields constituting a 400x800 m area located in Western Denmark, Northern Europe (Fig. 1). The barley fields were separated by a road and surrounded partly by other barley fields and partly by conifer forest. The upper 0.25 m of the soil profile consisted of homogeneous sandy loam and coarse sand with a relatively poor capacity to retain water was found below. A soil moisture content of 26% was measured at field capacity, and the soil porosity ranged between 0.35 and 0.40. The site was located in marine-influenced temperate climate with a mean

annual temperature of 8.2 °C and mean annual precipitation of 990 mm. Cloudy conditions are frequent in this area, with 1727 h of detected sunshine in 2014 (Cappelen, 2015).

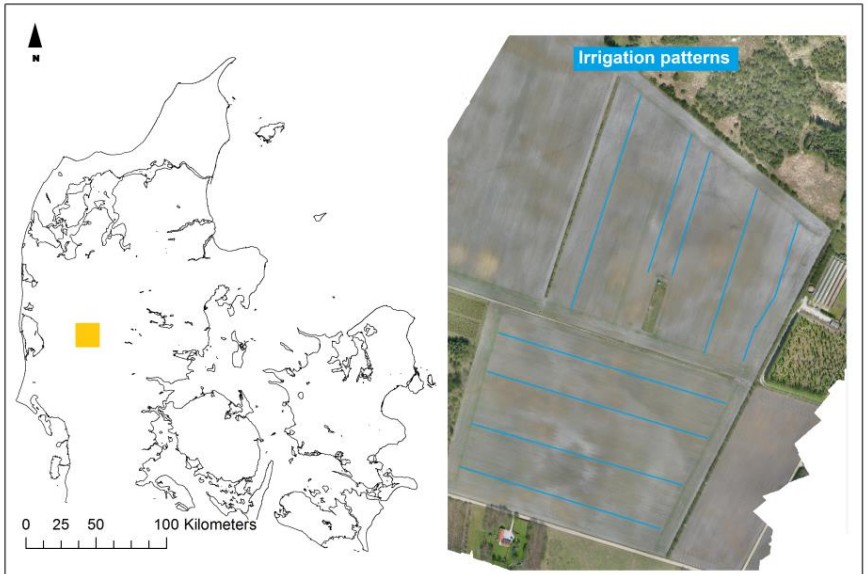

**Figure 1: The two barley fields located in Western Denmark (orange square: 56.037644°N, 9.159383°E). Right hand side shows an**

**orthophoto of the two barley fields. Blue lines show tramlines in which irrigation guns were placed.**

The fields were routinely irrigated according to normal practice on sandy soils. In 2014, irrigation was applied approx. on May 23, May 29, June 5, June15, and July 2. The irrigation of both fields takes approx. two days, and 25 mm of water was applied on each occasion using a travelling irrigation gun. Barley was sown on March 14 2014 and harvested on August 22 2014. Data were collected on three dates, April 22, June 18 and July 2, and the growth stages of the barley were, according to Tottman (1987), 13, 61 and 69, respectively. On April 22 the barley had approx. three unfolded leaves and a height of 0.08 m. On June 18 the barley was in the early stages of anthesis and 0.95 m tall, while on July 2 the anthesis was complete and the crops were 1.1 m tall.

## 2.2 WDI index

The WDI index is defined in Moran et al. (1994) as:

$$WDI = 1 - \frac{\lambda E_{act}}{\lambda E_{pot}} \quad = \quad \frac{(T_S - T_A)_{min} - (T_S - T_A)_{mes}}{(T_S - T_A)_{min} - (T_S - T_A)_{max}}, \tag{1}$$

where $\lambda E_{act}$ is the actual latent heat flux density (W m$^{-2}$), $\lambda E_{pot}$ is the potential latent heat flux density (W m$^{-2}$), $T_S$ is the composite LST (°C), $T_A$ is the air temperature (°C) and the subscripts 'min', 'max' and 'mes' refer to minimum, maximum and measured. Theoretical upper and lower limits of surface-air temperature differences (($T_S$-$T_A$)$_{max}$ and ($T_S$-$T_A$)$_{min}$) are constrained by the fraction of canopy cover. Upper and lower $T_S$-$T_A$ limits for a given meteorological forcing can be expressed through the surface energy balance, and measured values of the surface-air temperature differences (($T_S$-$T_A$)$_{mes}$) can thus be related to the theoretical $T_S$-$T_A$ span, and thereby to the actual to potential evapotranspiration ratio (Moran et al., 1994). ($T_S$-$T_A$)$_{max}$ and ($T_S$-$T_A$)$_{min}$ can be determined with the Vegetation Index Trapezoid (VIT) approach for a specific crop growth stage (Moran et al., 1994). Instead of depending on a single span of upper and lower $T_S$-$T_A$ values, the VIT approach thus considers upper and lower $T_S$-$T_A$ limits for full-cover green canopy, bare soil and all the stages in between. Four corresponding $T_S$-$T_A$ and VI values can be computed representing conditions with 1) well-watered full-cover vegetation 2) water stressed full-cover vegetation 3) saturated bare soil, and 4) dry bare soil. When the four extreme water- and canopy cover conditions are plotted in a 2D coordinate system with $T_S$-$T_A$ on the x-axis and VI on the y-axis a trapezoid shape appears, similar to the shape shown in see Fig. 2 after Moran et al. (1994).

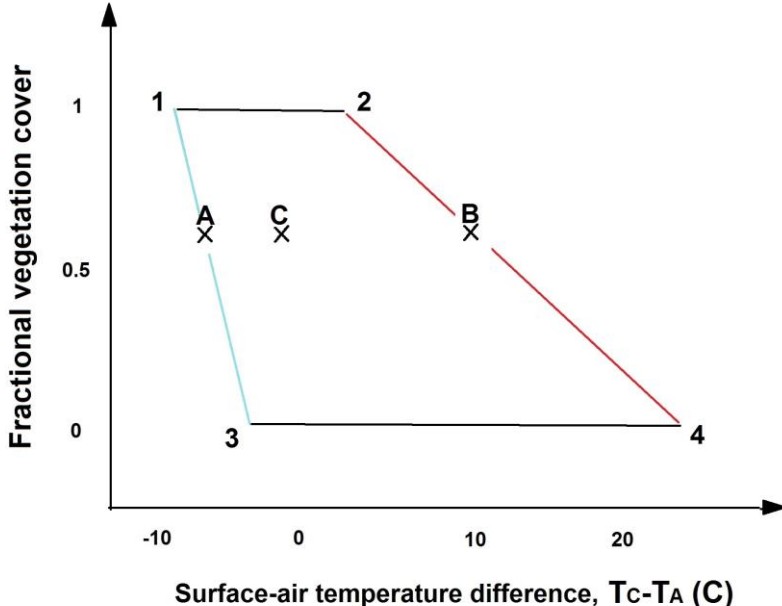

**Figure 2: Illustration of a standard shape of the trapezoid used for the VIT index calculations formed by four hypothetical vertices representing the extreme water- and canopy conditions: 1), 2), 3) and 4). 'C' represents corresponding values of measured $T_S$-$T_A$ and vegetation cover, 'A' represents theoretically calculated $(T_S$-$T_A)_{min}$, and 'B' represents theoretically calculated $(T_S$-$T_A)_{max}$ for the canopy fraction measured at 'C' (after Moran et al. (1994)).**

Corresponding $T_S$-$T_A$ and VI values calculated for the four extreme water- and canopy cover conditions are highlighted with numbers at the trapezoid corners. 'C' represents a measured value of surface-air temperature difference $((T_S$-$T_A)_{mes})$ and its corresponding measured vegetation cover. 'A' represents theoretical $(T_S$-$T_A)_{min}$ and 'B' represents theoretical $(T_S$-$T_A)_{max}$ for the specific fraction of canopy cover measured in C. Graphically, the ratio of CB to AB distances represent the ratio of actual to potential evapotranspiration. WDI can thus be formulated as WDI = 1- $\lambda E_{act}/\lambda E_{pot}$ = AC/AB = $\frac{(T_S-T_A)_{min}-(T_S-T_A)_{mes}}{(T_S-T_A)_{min}-(T_S-T_A)_{max}}$, see Eq. (1). Computations of $(T_S$-$T_A)_{min}$ and $(T_S$-$T_A)_{max}$ for the extreme water- and canopy conditions are based on the surface energy balance for a crop canopy:

$$R_n = G + H + \lambda E, \tag{2}$$

where $R_n$ is the net radiation, G is the soil heat flux, H is the sensible heat flux, and $\lambda E$ is the latent heat flux density (all in W m$^{-2}$). H and $\lambda E$ can be expressed according to Monteith (1973) and Monteith and Szeicz (1962) as:

$$H = C_v \left(\frac{T_S-T_A}{r_a}\right), \tag{3}$$

$$\lambda E = C_v \left(\frac{VPD}{\gamma(r_a+r_c)}\right), \tag{4}$$

where $C_v$ is the volumetric heat capacity of air (J °C$^{-1}$), $T_S$ is the LST (°C), VPD is the vapor pressure deficit of air (kPa), $\gamma$ is the psychometric constant (kPa °C$^{-1}$), and $r_a$ is the aerodynamic resistance (m s$^{-1}$) and $r_c$ the canopy resistance (s m$^{-1}$) to

vapor transport. Defining $\Delta$ as the slope of the saturated vapor pressure-temperature relation, $(e_C{}^*-e_A{}^*/T_C-T_A)$ (kPa °C$^{-1}$), Eq. (2), (3) and (4) can be combined and solved for $T_S-T_A$ as in Jackson et al. (1981):

$$(T_S - T_A) = \left[\frac{r_a(R_n-G)}{C_v}\right]\left[\frac{\gamma\left(1+\frac{r_c}{r_a}\right)}{\{\Delta+\gamma\left(1+\frac{r_c}{r_a}\right)\}}\right] - \left[\frac{VPD}{\{\Delta+\gamma\left(1+\frac{r_c}{r_a}\right)\}}\right], \tag{5}$$

When $T_S$-$T_A$ are expressed for conditions with well-watered full-cover vegetation (extreme condition 1) the canopy resistance is set to be valid for crop transpiration at potential rate ($r_{cp}$):

$$(T_S - T_A)_1 = \left[\frac{r_a(R_n-G)}{C_v}\right]\left[\frac{\gamma\left(1+\frac{r_{cp}}{r_a}\right)}{\{\Delta+\gamma\left(1+\frac{r_{cp}}{r_a}\right)\}}\right] - \left[\frac{VPD}{\{\Delta+\gamma\left(1+\frac{r_{cp}}{r_a}\right)\}}\right], \tag{6}$$

For water stressed full-cover vegetation (extreme condition 2), $r_c$ is replaced with maximum canopy resistance representing conditions with almost no transpiration ($r_{ex}$):

$$(T_S - T_A)_2 = \left[\frac{r_a(R_n-G)}{C_v}\right]\left[\frac{\gamma\left(1+\frac{r_{ex}}{r_a}\right)}{\{\Delta+\gamma\left(1+\frac{r_{ex}}{r_a}\right)\}}\right] - \left[\frac{VPD}{\{\Delta+\gamma\left(1+\frac{r_{ex}}{r_a}\right)\}}\right], \tag{7}$$

For saturated bare soil (extreme condition 3), canopy resistance is zero ($r_c$=0) and the resistance to heat transfer in the boundary layer immediately above the soil surface ($r_S$) defined by Shuttleworth and Wallace (1985) [(Norman et al., 1995)], is added to the aerodynamic resistance:

$$(T_S - T_A)_3 = \left[\frac{(r_a+r_S)(R_n-G)}{C_v}\right]\left[\frac{\gamma}{\{\Delta+\gamma\}}\right] - \left[\frac{VPD}{\{\Delta+\gamma\}}\right], \tag{8}$$

For dry bare soil (extreme condition 4), canopy resistance is set to infinity ($r_c$=∞) and the soil resistance remains in the equation:

$$(T_S - T_A)_4 = \left[\frac{(r_a+r_S)(R_n-G)}{C_v}\right], \tag{9}$$

The WDI index resembles the two-source models (Norman et al., 1995; Shuttleworth and Wallace, 1985) since it distinguishes between fluxes from bare soil and vegetation. Hoffmann et al. (2016) have shown that the two-source energy models presented in Norman et al. (1995) and Norman et al. (2000) are applicable in overcast weather, and we therefore hypothesized that the WDI index (as opposed to the CWSI index) would be applicable in overcast weather as well. Further, Hoffmann et al. (2016) showed that subtracting air and LST obtained one hour after sunrise, from air and LST obtained at midday (the dual-temperature-difference approach (Norman et al., 2000)) improves estimates of surface energy balance-components based on remotely sensed thermal data. Temperatures obtained one hour after sunrise represent a starting point for the daily temperature development, and adding these additional observations accounts for any offset between collected air and LST caused by, for example, atmospheric effects (Norman et al., 2000). The same procedure was therefore applied in this study, so that for conditions with well-watered full-cover vegetation the computations become:

$$(T_S - T_A)_1 = (T_{S,i} - T_{A,i}) - (T_{S,0} - T_{A,0}) = \left[\frac{r_{a,i}(R_n-G)_i}{C_v}\right]\left[\frac{\gamma\left(1+\frac{r_{cp,i}}{r_{a,i}}\right)}{\left\{\Delta+\gamma\left(1+\frac{r_{cp,i}}{r_{a,i}}\right)\right\}}\right] - \left[\frac{VPD_i}{\left\{\Delta+\gamma\left(1+\frac{r_{cp,i}}{r_{a,i}}\right)\right\}}\right] - \left[\frac{r_{a,0}(R_n-G)_0}{C_v}\right]\left[\frac{\gamma\left(1+\frac{r_{cp,0}}{r_{a,0}}\right)}{\left\{\Delta+\gamma\left(1+\frac{r_{cp,0}}{r_{a,0}}\right)\right\}}\right] -$$

$$\left[\frac{VPD_0}{\left\{\Delta+\gamma\left(1+\frac{r_{cp,0}}{r_{a,0}}\right)\right\}}\right], \tag{10}$$

The same procedure is applied to conditions $(T_S\text{-}T_A)_2$, $(T_S\text{-}T_A)_3$ and $(T_S\text{-}T_A)_4$.

The upper and lower VI values (y-values) are dependent on maximum and minimum values extracted from the vegetation index maps. The y-values have to span the values obtained from areas with green full canopy coverage and from areas with bare soil and ripe crops. In this study the NGRDI is used as VI.

## 2.3 NGRDI Index

The NGRDI index is computed according to Hunt Jr et al. (2005) as:

$$NGRDI = (Green\ DN - Red\ DN)\ /\ (Green\ DN + Red\ DN), \tag{11}$$

where Green DN and Red DN are digital numbers recorded in the green and red bands, respectively. The difference between the green and the red band (numerator) reveals the canopy-soil fraction, while the sum of the green and the red band (denominator) normalizes the index and accounts for variations of light intensity caused by variations in exposure time. The NGRDI index spans from -1 to 1, where positive numbers tending towards 1 represent more green vegetation and negative values represent bare soil. Hunt Jr et al. (2005) observed a saturation of NGRDI in ripening corn, but because barley has a lower biomass than corn, saturation of the NGRDI index was not regarded as an issue in this study. Based on their color, we assumed that ripening barley and bare soil had a similar green-red DN response, and therefore that ripe barley and bare soil had the same evaporation response. This is an approximation however, since ripe crops cease photosynthesis, the evaporation process will resemble the one from bare soil. Further, even if the physical responses in ripe crops and bare soil are not the same, the VIT approach requires comparably higher $T_S$-$T_A$ measurements of ripe crops (and bare soil) than of transpiring crops before its results are translated into water stressed regions (see Sect. 2.2 and shape of trapezoid in Fig. 2). We hypothesized that this mechanism would produce the desired response in WDI maps; areas with ripe or prematurely ripe crops should ideally appear as less water stressed in irrigation maps, regardless of the soil water content, as areas with ripe crops (which are in their final developmental stage ) do not need the same volume of irrigation as green crops.

## 2.4 Trapezoid computations

Computations of $T_S$-$T_A$ in extreme soil water and canopy conditions on the three campaign days are based on the data shown in Table 1. Vapor pressure deficit (VPD) was calculated with measurements from a Campbell Scientific HMP45C Temperature and Relative Humidity probe. Soil heat flux (G) was measured with heat flux plates (Hukseflux Thermal Sensor model HFP01), and net radiation ($R_n$) was measured with a Hukseflux four-component radiation sensor (model NR01). The aerodynamic resistance ($r_a$) was calculated after Allen et al. (1998) and $r_S$ after Norman et al. (2000). $r_{cp}$ and $r_{ex}$ were

obtained as in Monteith (1973) where $r_{cp}$ = $r_{sm}$/LAI and $r_{ex}$ =$r_{sx}$/LAI. Values for minimum stomatal resistance ($r_{sm}$) and maximum stomatal resistance ($r_{sx}$) were 25 s m$^{-1}$ and 1000 s m$^{-1}$ respectively, as suggested in the same paper. LAI was collected with a plant canopy analyzer (LAI2000) and extrapolated to the three UAV data collection dates, as described in Hoffmann et al. (2016). Volumetric heat capacity ($C_v$), psychrometric constant ($\gamma$) and the slope of the saturated vapor

pressure-temperature relation ($\Delta$) are kept fixed for all three dates with values of 1200 J °C$^{-1}$, 0.066 kPa °C$^{-1}$ and 0.1098 kPa °C$^{-1}$.

Maximum NGRDI values of 0.25 were detected in June 18 and minimum values of -0.1 were detected on July 2. Upper and lower y-values in trapezoids have to cover at least the max and min values; the span establishes how sensitive the WDI index is to the measured NGRDI values. Since the aim is for the $T_S$-$T_A$ and NGRDI maps to affect WDI maps equally, after an

empirical evaluation, an upper level of 0.4 and a lower level of -0.3 were chosen for all three trapezoids. The same upper and lower NGRDI values were chosen for all three trapezoids to enable comparison of the resulting WDI maps. Final trapezoids with computed $(T_S\text{-}T_A)_{max}$ and $(T_S\text{-}T_A)_{min}$, along with measured $T_S$-$T_A$ and NGRDI, are shown in Appendix A.

## 2.5 RGB and Thermal UAV data

RGB and thermal data were collected with a fixed-wing UAV on three occasions during 2014: April 22, June 18 and July 2.

The lithium-battery-driven UAV was in the air for approx. 20 min on each occasion. The thermal camera was an Optris PI 450 camera that detects infrared energy in the 7.5–13 µm electromagnetic spectrum and computes temperature images of 382 x 288 pixels based on an emissivity of unity. Measurements are accurate to within ± 2 °C or ±2 % at an ambient temperature of 23 °C. The thermal camera is un-cooled and, being 0.32 kg, mountable on a UAV with a wingspan of 2 m (Fig. 3). UAV flights with the thermal camera were conducted at times when there was consistent cloud cover to ensure minimal variation

in radiance; however, small changes in cloud cover thickness did occur during the flights. Differences in detected thermal energy resulting from shaded and sunlit surfaces might have occurred on June 18, as the cloud cover that day was very thin. On April 22 and July 2 the clouds were sufficiently thick to avoid anomalies of the kind that may introduce errors in the subsequent data processing.

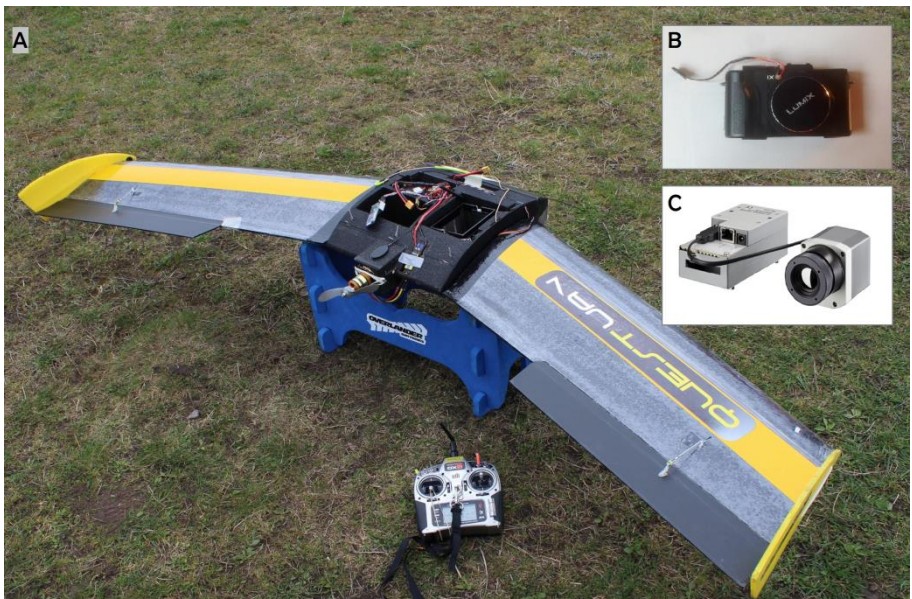

**Figure 3: (a) The fixed-wing QuestUAV with a wingspan of 2 meters used to collect data. (b) The digital Panasonic DMC-LX5 Lumix camera with installed trigger cable to ensure link between the autopilot GPS positions and GPS positions of triggered images. (c) The Optris PI 450 thermal infrared camera.**

The thermal camera was triggered by its internal auto-trigger and each image was assigned with GPS coordinates, as described in Hoffmann et al. (2016), by comparing the timestamp on images with the timestamp and GPS position extracted from the autopilot log file (created by autopilot software: SkyCircuit Ltd SC2). Thermal images were converted into unsigned 16-bit data to enable stitching of images in Agisoft PhotoScan software (Professional Edition version 1.0.4). The RGB data were collected with a digital Panasonic DMC-LX5 Lumix camera in which exposure times are adjusted

automatically with a built-in light intensity meter. The RGB camera was triggered using pre-defined settings in the SkyCircuit software, and georeferencing was conducted directly in the autopilot log file, as each triggered image was marked next to its corresponding GPS position. No conversion of RGB images was needed for the PhotoScan framework. A flying height of 90 m above ground allowed resolutions of 0.1 m pixel$^{-1}$ for the RGB data and 0.2 m pixel$^{-1}$ for the thermal data. Both RGB and thermal images were triggered every two seconds and the final number of approx. 400 images per flight were

aligned and stitched together using a mean value composition. PhotoScan uses the Structure-from-Motion Bundle Adjustment technique (Bolles et al., 1987; Triggs et al., 1999) in its image processing.

Half-hourly averages of air temperature measurements ($T_A$) made from a meteorological tower at the site were collected during UAV flights and subtracted from the stitched thermal images. Maps of the surface-air temperature difference (($T_S$-$T_A)_{mes}$) were used in further WDI calculations, as described in Sect. 2.2. Extraction of green and red bands from the RGB

imagery, along with the NGRDI computations, were performed with ImageJ software (Schindelin et al., 2015; Schneider et al., 2012). NGRDI and ($T_S$-$T_A)_{mes}$ maps were stacked to ensure same pixel size and the resulting resolution was 0.25 m. WDI

calculations were performed for each set of corresponding NGRDI and $(T_S-T_A)_{mes}$ pixels using MatLab (MATLAB and Statistics Toolbox Release 2012b, The MathWorks, Inc., Natick, Massachusetts, United States).

## 2.6 Validation data

### 2.6.1 Data for NGRDI validation

NGRDI maps were compared with UAV-RGB orthophotos created in PhotoScan. The NGRDI map from June 18 was also validated against measurements of Leaf Area Index (LAI) and against near-field remote sensing measurements of NDVI. LAI measurements were obtained on June 18 with a plant canopy analyzer: LAI2000 at six locations selected according to empirically confirmed different crop yields from previous years. LAI measurements were repeated four times at each location and a mean represented the final LAI value for each specific location. NDVI data were collected on June 22 with a MobilLas mounted on a tractor with three sensors: a near-infrared laser range finder (AccuRange 4000, Acuity Research Inc., CA, USA), a two-band radiometer (Crop Circle ACS-210, Holland Scientific Inc., NE, USA) and a global positioning system (GPS 16, Garmin International Inc., KS, USA). The two-band radiometer was an active canopy sensor providing its own illumination, and measurements were made independently of solar radiation. NDVI was calculated as in Carlson and Ripley (1997) with reflectance measurements made at near-infrared (780 nm) and red (670 nm) frequencies. Measurements were made at a 50° off-nadir angle. The GPS position was received at a rate of 1 Hz and a full measurement was recorded approximately every 1.6 seconds. A measurement was recorded for approx. every 2 m travelled along the tramlines. A total of about 1300 measuring points were collected from both fields.

### 2.6.2 Data for temperature validation

The surface-air temperature difference maps were compared with measurements from time domain reflectometry (TDR) probes providing information on volumetric soil water content. Six TDR probes were placed at the same locations as those for the LAI measurements: three in the field north of the road and three south of the road. Measurements were collected on nine days during the growing season, 2014: March 27, April 24, May 8, May 13, May 27, June 4, June 13, June 26 and July 17. The TDR100 instrument (Campbell Scientific, Logan, UT, UAS) was deployed as a probe system with a central hub connecting the six individual probes. The central hub ensures that measurements can be taken without disturbing the vegetation and soil where probes are installed (Thomsen, 2006). The probe rods measured soil water content representing the first 0.2 m and 0.5 m of the soil profile. Three repeat measurements were made at each location, and volumetric soil water contents were calculated using the apparent dielectric constant, as in Topp et al. (1980).

### 2.6.3 Data for WDI validation

Mean values from WDI maps were validated with measured stress values computed as $1 - \frac{\lambda E_{eddy}}{Rn-G}$, where $\lambda E_{eddy}$ is the latent heat flux measured with an eddy covariance system consisting of a Gill R3-50 sonic anemometer and a LI-7500 open path

infrared gas analyser located between the two barley fields. Eddy covariance data were processed using EddyPro version 4.1. $R_n$ is the net radiation measured with the Hukseflux four-component radiator presented in Sect. 3.1, and G is the soil heat flux measured with Hukseflux heat flux plates. $\lambda E_{eddy}$ thus represents $\lambda E_{act}$ and $R_n$-G represents $\lambda E_{pot}$ in this stress value, see Eq. (1). Further, WDI maps were compared with the CWSI index, which was computed using two different approaches: the limit approach (derived from Jackson et al. (1981), Eq. (10)) and the resistance approach (derived from Jackson et al. (1981), Eq. (6) and (7)). In the resistance approach, actual canopy resistance was required, and in this study it was obtained according to Herbst et al. (2007).

## 3 Results and Discussion

Maps of NGRDI, $T_S$-$T_A$ and WDI, together with orthophotos from the RGB camera, are shown in Fig. 4-6 (A-D), for April 22, June 18 and July 2.

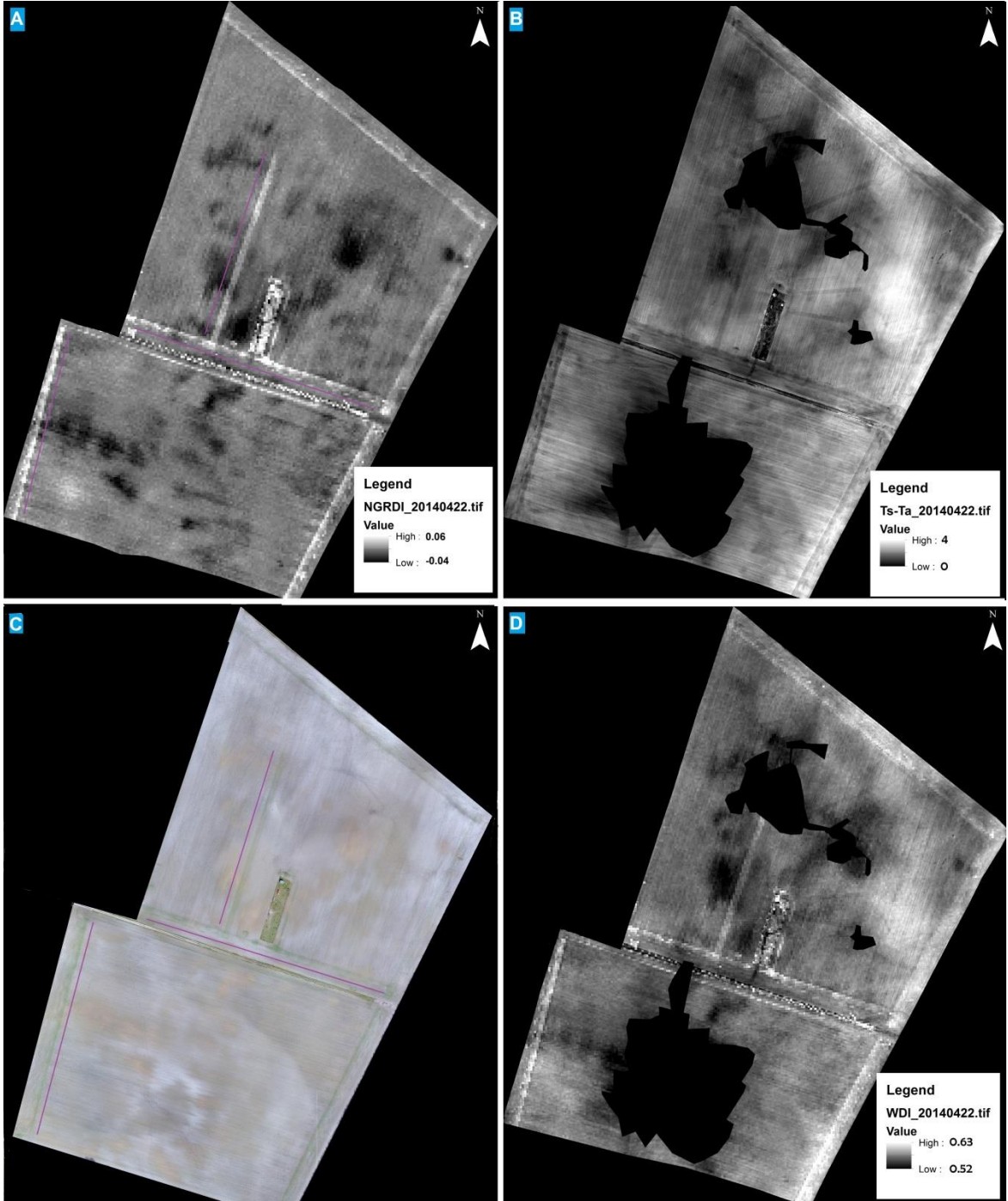

**Figure 4: Imagery and indices from April 22. A) NGRDI, B) $T_S$-$T_A$, C) orthophoto (RGB data) and D) WDI. Black areas (no data) within the field on thermal maps B) and D) are due to insufficient texture in images collected in these areas, which meant that the images could not be aligned with the Structure-from-Motion algorithm. The RGB imagery do not suffer from insufficient texture as these have a higher spatial resolution compared to the thermal imagery. Purple lines are placed next to green stripes of vegetation.**

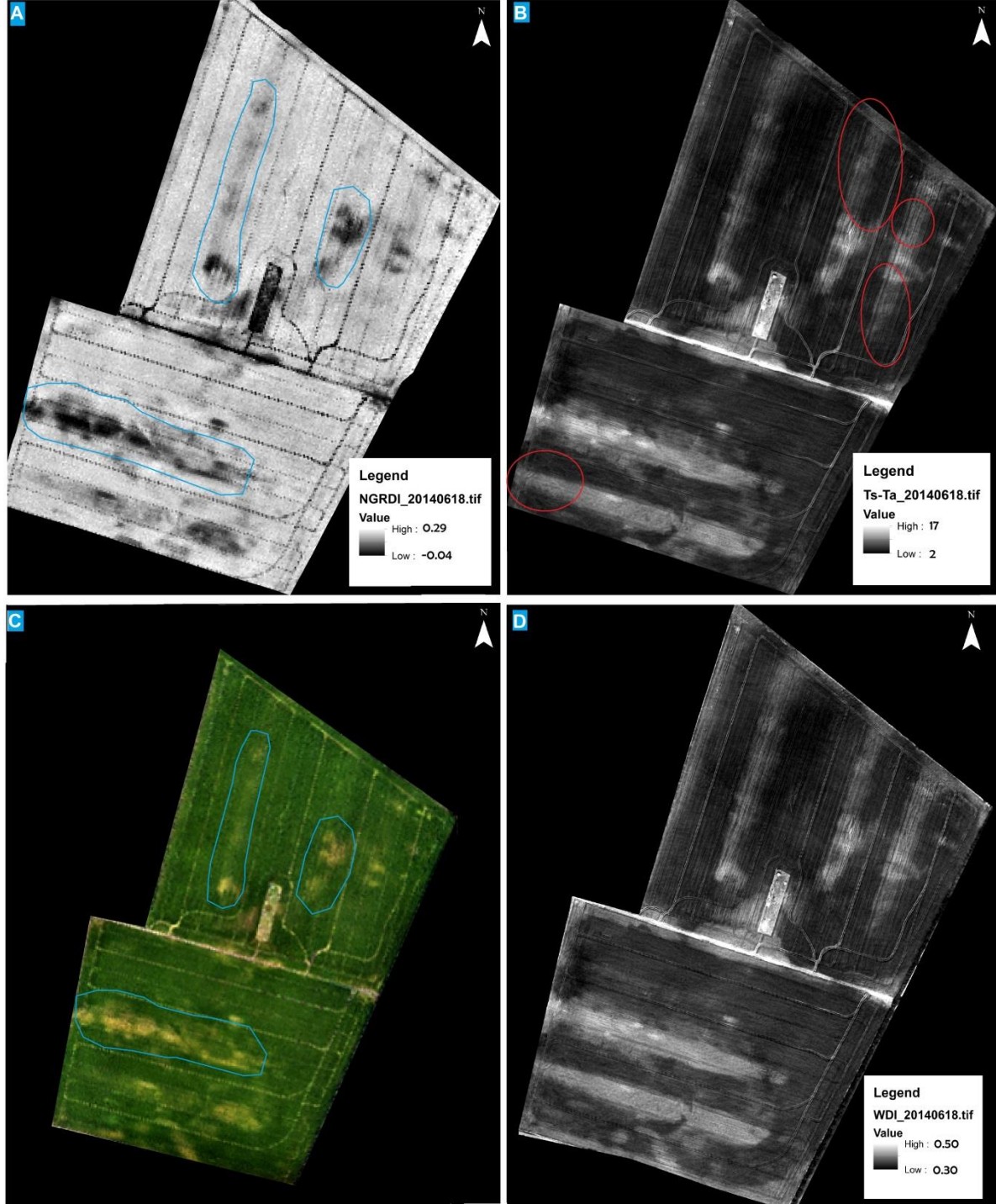

**Figure 5: Imagery and indices from June 18. A) NGRDI, B) $T_S$-$T_A$, C) orthophoto (RGB data) and D) WDI. Red circles in B) are explained in Sect. 4.2. Areas highlighted with blue are examples of locations with ripe crops.**

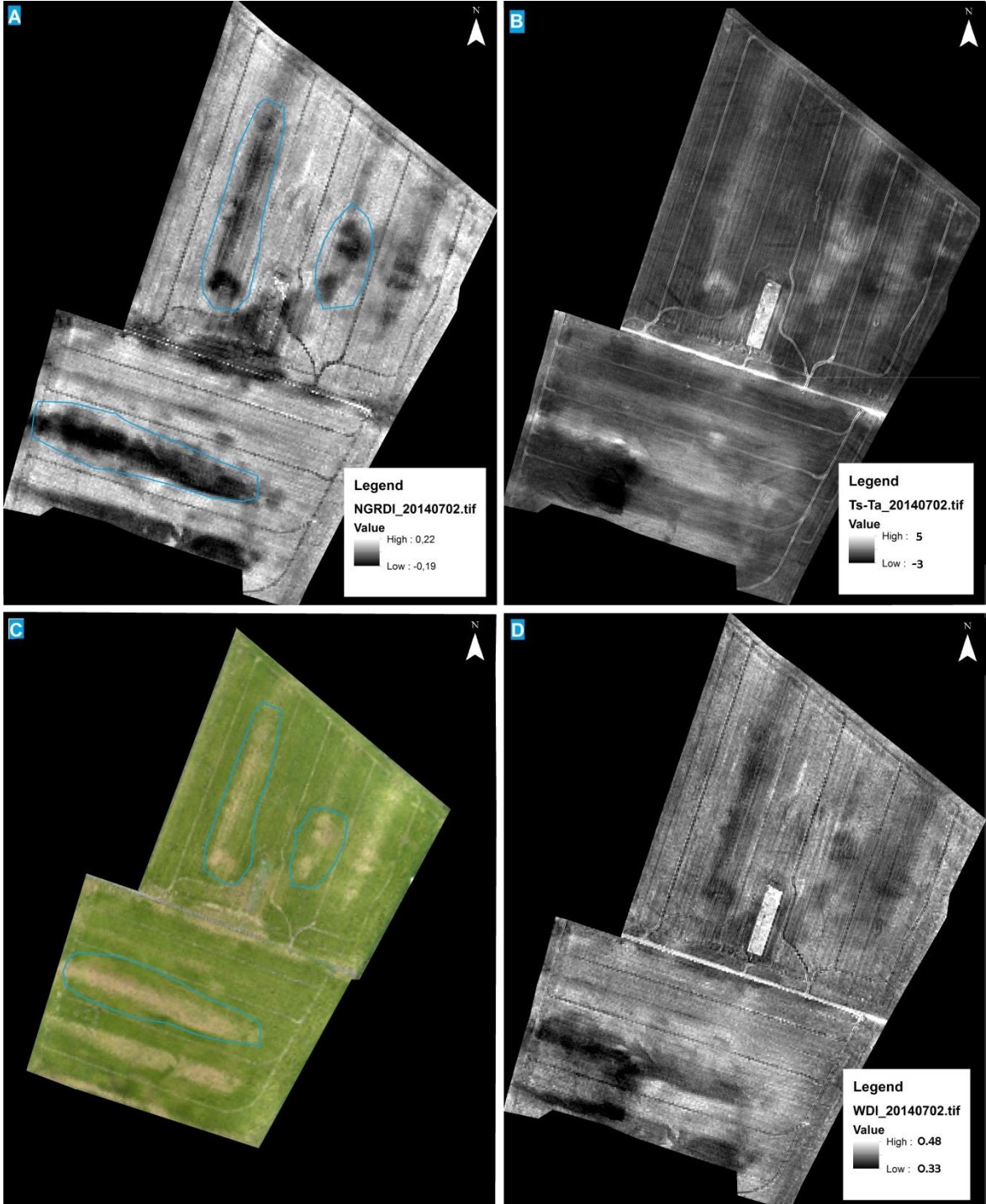

**Figure 6: Imagery and indices from July 2. A) NGRDI, B) $T_S$-$T_A$, C) orthophoto (RGB data) and D) WDI. Areas highlighted with blue are examples of locations with ripe crops.**

### 3.1 NGRDI maps

Visual comparison of NGRDI maps and corresponding orthophotos (Figs. 4-6, A and C) confirms that the NGRDI index can be used to quantify greenness from the RGB data. This is observed relative to soil (Fig. 4) and relative to yellowing barley (Fig. 5 and 6). The green stripes along the edges of the fields in the orthophoto and a northeast-southwest orientated stripe of green vegetation in the field north of the gravel road in Fig. 4C are translated into brighter areas in Fig. 4A (purple lines), indicating more green vegetation and a higher canopy cover. Comparison of map 5A with 5C and 6A to 6C shows that the NGRDI index is also able to quantify the degree of crop ripeness. Areas with ripe crops attain lower (darker) NGRDI values (areas highlighted with blue). The same NGRDI response is observed in areas with bare soil (Fig. 4A), and therefore the NGRDI responses to ripe crops and bare soil are indeed similar.

The NGRDI map from June 18 was compared with approx. 1300 spatially distributed measurements of NDVI from June 22, and a correlation coefficient (r) of 0.56 was achieved (p<0.001) (Table 2). See Appendix B for NDVI-NGRDI plot. The offset of four days between NDVI and NGRDI measurements was expected to introduce small differences in the detected greenness in areas where vegetation is near senescence. The correlation might have been stronger had the dates corresponded. However, it suggests a medium-strong association.

High LAI values were measured in areas with high NGRDI values and likewise with low LAI and NGRDI values. A correlation coefficient of 0.85 was achieved between these two parameters. On June 18, low LAI measurements were located in areas with ripe crops (see Appendix C). Yellow crops on this date were presumably pre-maturely ripe and therefore smaller. Correlations between the NGRDI map and LAI and NDVI measurements bode well for the quality of collected RGB data and for the NGRDI as greenness index. A correlation has further been conducted between LAI and NDVI to test which of NGRDI and NDVI best represented the LAI measurements (see Table 2). The correlation coefficients are equally high. According to Knipling (1970), crop reflectivity response in the visible spectrum depends on leaf chlorophyll content and information from the visible bands thus enables a distinction between green and yellow crops. However, vegetation indices based on spectrally narrow bands collected with advanced sensors in the visible and near-infrared spectra are most commonly used when crop conditions are assessed (Baluja et al., 2012; Garcia-Ruiz et al., 2013; Lelong et al., 2008; Primicerio et al., 2012; Sugiura et al., 2005; Zarco-Tejada et al., 2012). Rasmussen et al. (2016) concludes that there is no difference in the ability to detect spectral crop response between consumer grade RGB cameras (broad bands) and multispectral sensors providing narrow-band information in visible and near-infrared spectra.

### 3.2 $T_S$-$T_A$ maps

$T_S$-$T_A$ maps for April 22, June 18 and July 2 are shown in Figs. 4B, 5B and 6B. Temperatures vary most in the June 18 map (see legend in 5B) which corresponds with the highest net radiation measured on this date (Table 1). Differences in soil water availability become more obvious in days with high available energy.

There was clear agreement between $T_S$-$T_A$ and TDR measurements (Table 3). The correlation coefficient spanned from -0.57

to -0.83 and thus indicated a strong negative correlation between soil water content and the surface-air temperature difference.

Appendix D shows the location of the six TDR probes on the $T_S$-$T_A$ map from June 18 (Fig. D1), along with a graph showing the soil water contents measured at the six TDR probes during the entire growing season (Fig. D2).

The shape of the elongated warmer areas in $T_S$-$T_A$ maps from June 18 and July 2 are unlikely to be due to soil properties or other natural occurring phenomena. Comparing the shape and placement of warmer areas to the placement of irrigation guns (Fig. 1) gives strong incitements to theorize a correlation between the two. The repeated placement of irrigation guns in tramlines has resulted in less water being applied in areas furthest away from the tramlines. Vegetation in these areas has less available water and thus sooner becomes water stressed and warmer compared to surrounding vegetation. Further, a

darker/colder area can be seen in the lower right corner of Fig. 6B. This is the location of an irrigation gun, applying water to the crops as the UAV is collecting data. The fact that irrigated areas appear colder provides confidence in the intra-field spatial variations that the collected LST reveal.

Comparison of Figs. 5B, 5C and 7C reveals that thermal UAV data contain more information than it is possible to see simply by looking at the RGB images i.e. more than what can be seen with the naked eye. White areas highlighted by red circles

(Fig. 5B) indicate higher temperatures and higher soil water deficits that cannot be detected in the orthophoto (Fig. 5C). Crops in the areas highlighted with red circles in the June 18 map become yellow by July 2 (Fig. 6C). The temperature data from June 18 thus predicts where crops on July 2 will become prematurely ripe as a consequence of being water stressed and receiving insufficient irrigation.

### 3.3 WDI maps

Figures 4-6 A, B and D show that variations in both $T_S$-$T_A$ and NGRDI maps are reflected in the WDI maps. This can most clearly be seen on April 22 (Fig. 4 A, B and D) where variations in NGRDI and $T_S$-$T_A$ maps deviate most from each other. On June 18, areas with high $T_S$-$T_A$ and green vegetation (high NGRDI values) are translated into areas with higher water deficits (Fig. 6D). The same areas have high $T_S$-$T_A$ on July 2. However, the increasing incidence of ripe/prematurely ripe crops in the same areas now translates into lower WDI values. When areas are detected as those with ripe/prematurely ripe

crops, the $T_S$-$T_A$ has to be higher in order to result in a WDI index that predicts stress (see trapezoid explanation in Sect. 2.2). It is more difficult for WDI to indicate stress in senesced crops with low transpiration. This confirms the assumed added utility of the WDI index when the NGRDI index is extended and applied in the late growing season. In practice, ripe/prematurely ripe crops do not require any further irrigation as they have reached their final developmental stage.

Mean values and standard deviations of the WDI maps are shown in Table 4, along with computed CWSI indices and

measured stress values (1- $\lambda E_{eddy}$ / ($R_n$-G)). The mean values of WDI and the measured stress values correspond well on all three days. The added utility and accuracy of the WDI is supported by comparison with the CWSI values. With one exception, the CWSI indices overestimate the degree of stress in the barley fields compared to the measured stress values. The exception is the $CWSI_{res}$ on April 22, which is underestimated. This confirms the need for an alternative stress index to

CWSI in situations with partially vegetated fields and in fields with areas of ripe crops.

The WDI is highest in April and lowest in June and July. On April 22 the barley was in an early growth stage, so the root networks in the fields were not yet large nor deep. Further, the low LAI values on this date indicated that small fractions of the field of view were occupied by canopy. The LST collected on April 22 is thus an expression of soil water in the upper-most soil layer, which may dry rapidly. This led to the high WDI of 0.58 in April 22. WDI results from fields with low LAI and shallow root network are correct according to measured stress values, and thus also according to actual evapotranspiration. However, they only represent soil water availability in the top few centimeters of the soil and are therefore less representative of the deeper soil water content. The relatively low stress values of 0.40 and 0.41 obtained in June and July agree well with the repetitive irrigation of the fields initiated on May 23. The large variation in the WDI map from June 18 is due to the large amount of available energy detected on this day. Crops with insufficient soil water availability will heat up in accordance with the large amount of available energy. Variations in soil water will result in large variations in crop temperature.

The WDI index provides very accurate estimates of crop water status, and therefore the WDI maps provide precise irrigation maps. WDI maps are most valuable in the late growing season. At this stage the remotely sensed data represent plants' available water more sensitively than they do in the early growing season, where the majority of the remotely sensed data represent water availability in the top few cm of soil profile. The WDI index accommodates absolute stress values. However, there is a degree of empirical assessment associated with the determination of upper and lower VI values in constructing the trapezoid. The $T_S$-$T_A$ maps and the WDI maps have largely similar spatial variations, they reveal soil water deficits before they can be detected with the naked eye (Sect.4.2) and thus predict where crops will become prematurely ripe without sufficient irrigation. Using the $T_S$-$T_A$ maps as irrigation maps would be simpler, and therefore advantageous in practical applications on cultivated lands. However, $T_S$-$T_A$ maps do not account for, and adjust to, stress values in areas with ripe/prematurely ripe crops. Further, they only provide relative stress values, and so empirical assessments will also be required (and to a greater extent) where irrigation decisions are based on $T_S$-$T_A$ maps alone.

The added utility of WDI response in late growing seasons is dependent on the color change in ripening crops, and crops with color change that is similar to that in barley (green to yellow) will probably be suitable for beneficial monitoring with WDI applications. We therefore expect that the added utility of the WDI index will be valid for many types of cereals. However, further studies will need to be conducted to generalize WDI advantages.

CWSI-thresholds for stressed and not stressed crops are species-dependent (Feldhake et al., 1997), and so the same will be true for WDI-thresholds. WDI-thresholds, which determine the amount of irrigation needed, are beyond the scope of this paper; they will need to be analyzed in future studies.

**4 Conclusions**

In this study, the UAV-based WDI index was applied to barley fields in April, June and July (2014), to investigate whether crop water deficits at sub-field scale could be determined at different crop growth stages. Data from both the early and the late growing season were investigated to assess whether the WDI index has the unique potential to be applicable both when
5 the land surface is partly composed of bare soil and when crops on the land surface are senescing.

We found that the WDI maps determined accurate absolute water stress values and variations within the barley fields, in agreement with measured stress values from the Eddy Covariance tower, at different growth stages. This implies that the WDI index accounts for areas with ripe and prematurely ripe crops that no longer need large volumes of irrigation. The robustness of the WDI index during different growth stages emphasizes its added utility compared with the more commonly
10 used CWSI index. Further, the WDI index has the potential to become an efficient and powerful irrigation tool in areas where overcast weather is common. We also found that the surface-air temperature difference maps alone can predict where crops will become prematurely ripe with insufficient irrigation. The study also demonstrated that a lightweight UAV system, a consumer grade camera and an un-calibrated and un-cooled thermal camera can be combined to produce accurate maps of crop water stress. In this way, reliance on camera calibration and costly multispectral cameras can be reduced. The WDI
15 response in the late growing season was crop-color dependent, and studies applying the setup we have presented to other crop types are needed in order to confirm the general added utility of the WDI index.

**Appendix A**

Computed trapezoids for the three data collection days. Red line is $(T_S-T_A)_{max}$, light blue line is $(T_S-T_A)_{min}$ and dark blue dots are $(T_S-T_A)_{mes}$

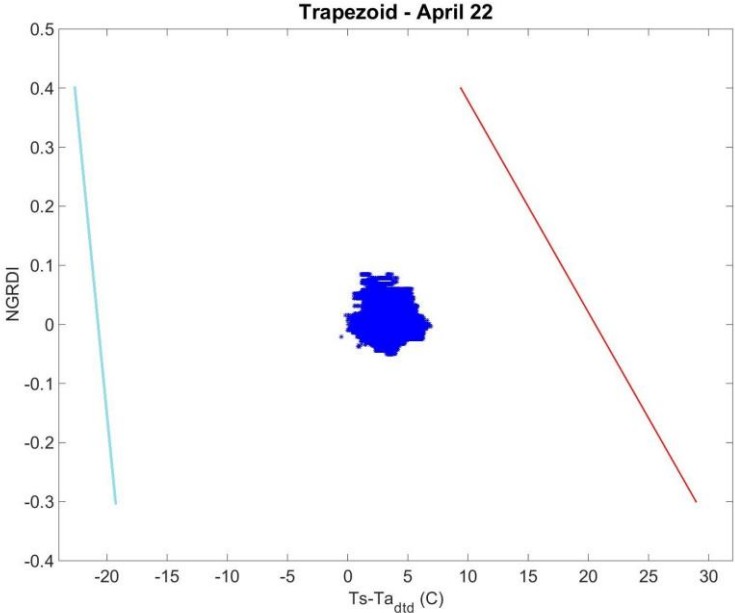

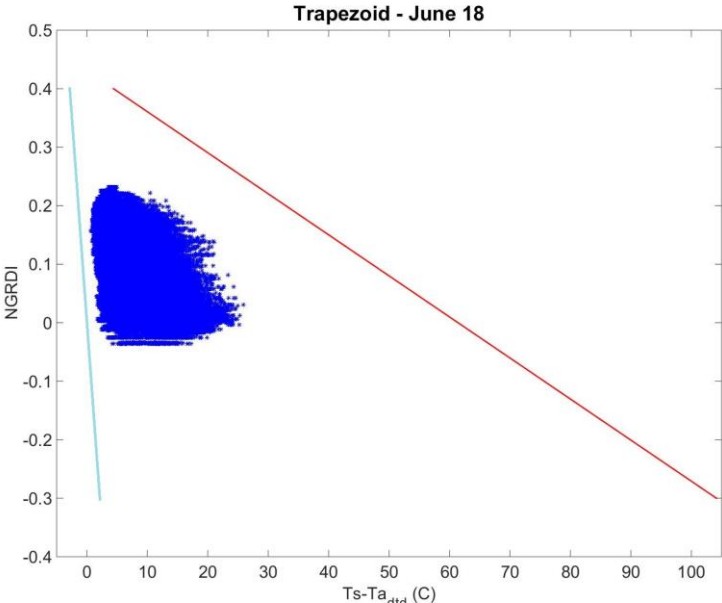

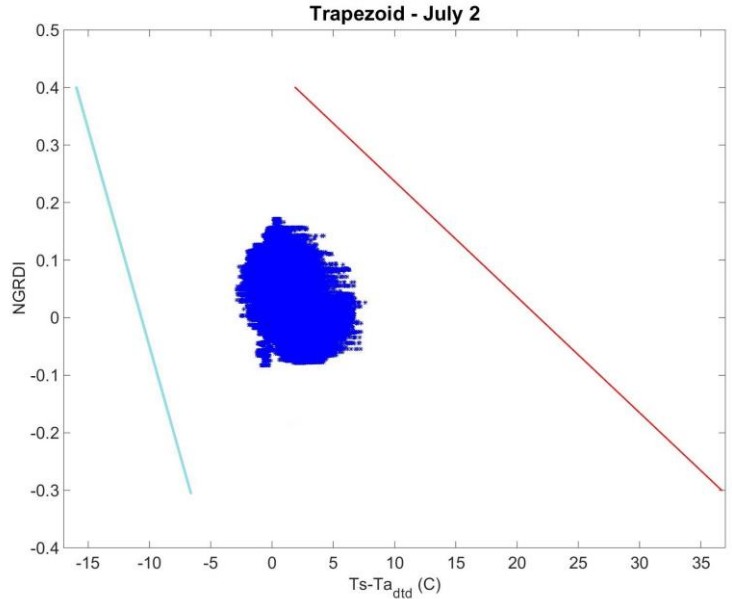

**Appendix B**

NGRDI from June 18 against NDVI from June 22. NDVI seems to saturate at high values of NGRDI.

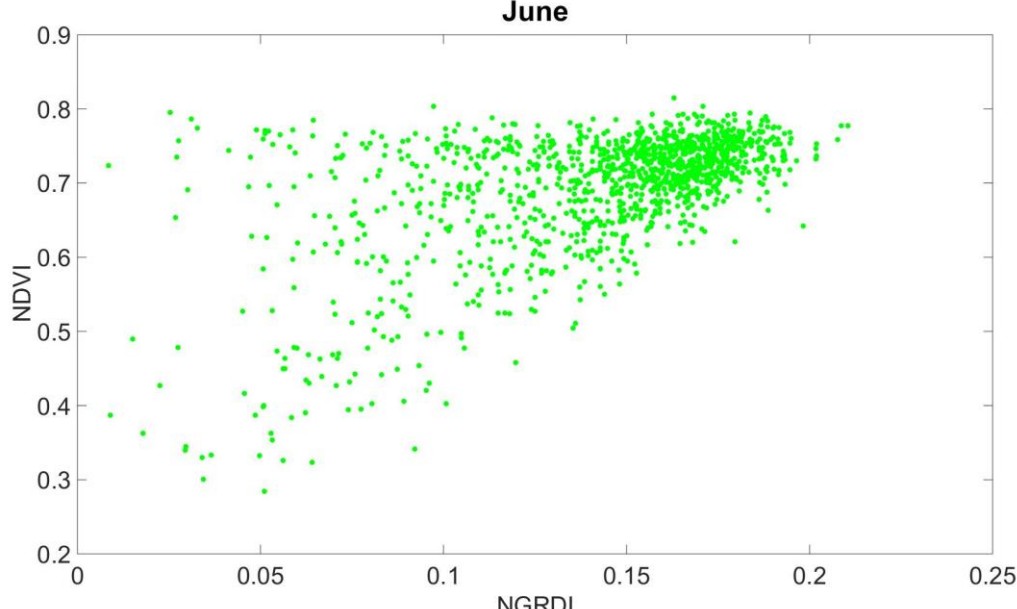

**Appendix C**

Location and size of LAI measurements on NGRDI map from June 18. Size of green stars indicates the magnitude of LAI, according to the legend. Darker areas on the background map represent low NGRDI values and areas with ripe/yellow crops. Brighter areas represent high NGRDI values with high green canopy cover. A correlation coefficient of 0.86 is obtained between NGRDI and LAI.

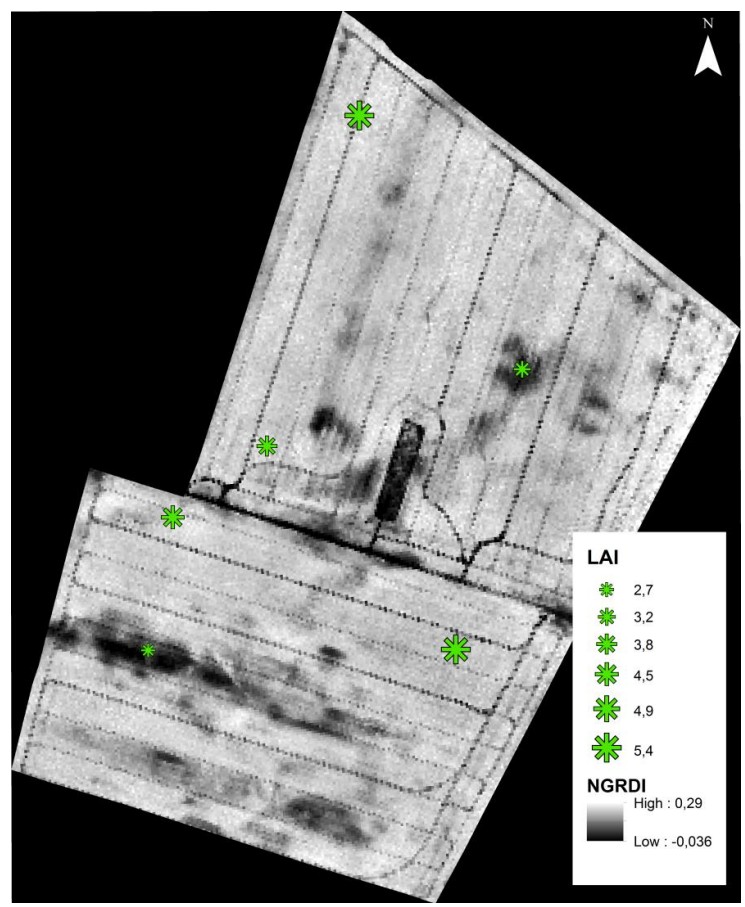

Figure D1: The location of the six TDR probes on the $T_S$-$T_A$ map from June 18.

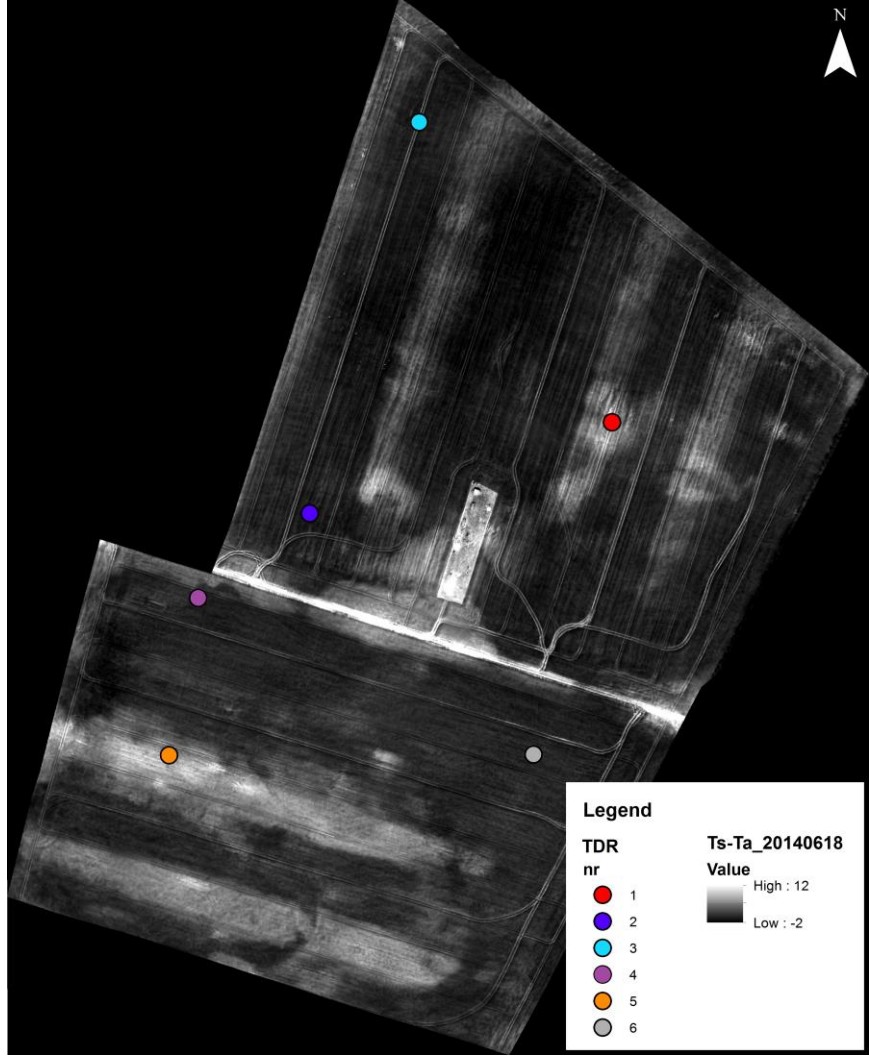

5    Figure D2: The soil water content measured at the six TDR probes during the growing season. Solid lines represent measurements at 0.2 m depth and semi-dashed lines represent 0.5 m depth. Black dashed lines indicate days of UAV data collection : April 22, June 18 and July 2.

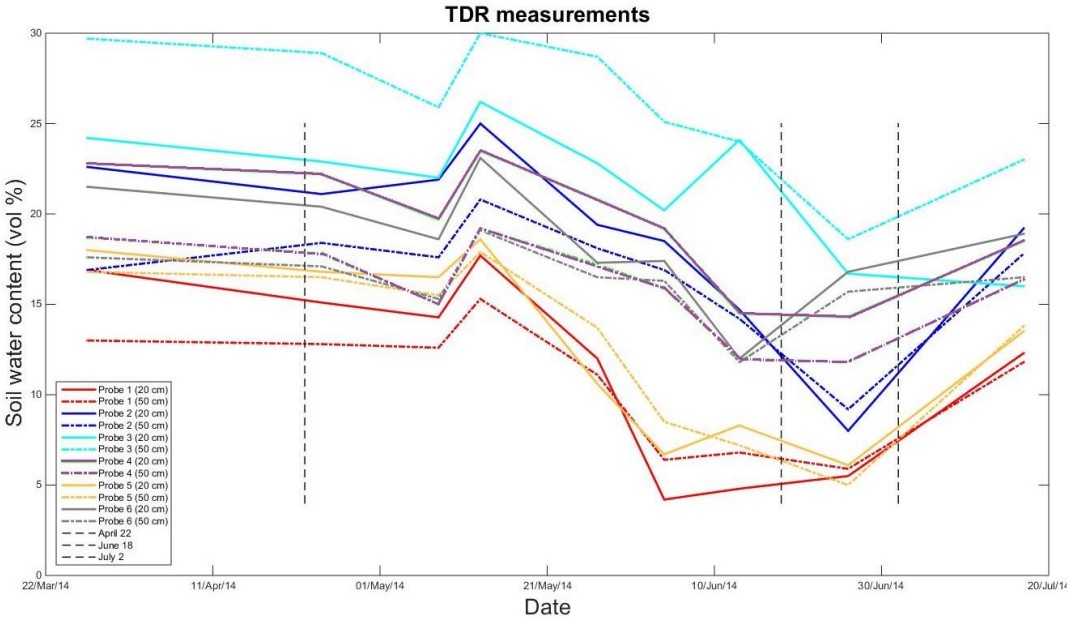

TDR probe 1 and 5 (red and orange) are located in areas with higher LST. Figure 9 shows that probes 1 and 5 measure lower soil water contents than the other four probes, especially on June 18 (black dashed lines in Fig. 9 show dates for UAV data collection). Comparison of the seasonal trend in A.2. with temperature maps from April 22 and July 2 (Figs. 4B and 6B) shows agreement between soil water content and temperature maps: the smaller temperature variance in the map from April 22 corresponds well with the small variability in soil water content among TDR probes in the same period. Similarly, the location of warmer areas in the June and July maps corresponds well with the lower soil water content seen at probes 1 and 5 during June and July.

**Author contribution:** Helene Hoffmann performed most of the data analysis, collected the UAV data and wrote the manuscript; Rasmus Jensen also collected UAV data and contributed to the analyses; Thomas Friborg and Hector Nieto contributed to the design of the study; Anton Thomsen collected validation data; Jesper Rasmussen contributed to the interpretation of data. All authors contributed to the editing of the manuscript.

**Acknowledgements:** This work was conducted within the Danish Hydrological Observatory, HOBE, and was funded by the Villum Foundation.

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

**Table 1: Data used to compute $T_A$-$T_S$ for the four extreme water and canopy conditions. Values before and after slashes (/) are data from midday and data from one hour after sunrise, respectively. Parameters are available energy ($R_n$ - G) where $R_n$ is net radiation and G is soil heat flux, vapor pressure deficit (VPD), aerodynamic resistance ($r_a$), resistance from air layer immediately above soil ($r_S$), canopy resistance at potential evapotranspiration ($r_{cp}$) and maximum canopy resistance, associated with nearly complete stomatal closure ($r_{ex}$) and leaf area index (LAI).**

| Date | $R_n$-G (W m$^{-2}$) | VPD (kPa) | $r_a$ (s m$^{-1}$) | $r_S$ (s m$^{-1}$) | $r_{cp}$ (s m$^{-1}$) | $r_{ex}$ (s m$^{-1}$) | LAI |
|---|---|---|---|---|---|---|---|
| April 22 | 154 / -33 | 8.2 / 2.9 | 74 / 87 | 110 | 22 | 1136 | 0.88 |
| June 18 | 579 / 8 | 7.0 / 0.5 | 34 / 56 | 200 | 6 | 248 | 4.03 |
| July 2 | 206 / -3 | 4.4 / 1.6 | 29 / 87 | 181 | 7 | 292 | 3.43 |

**Table 2: Correlation coefficients (r) between LAI and NGRDI, LAI and NDVI, and NDVI and NGRDI. The correlation between NDVI and NGRDI was based on 1300 measurements, while the correlations with LAI were based on six measurements. NDVI was interpolated using the Natural Neighbor technique.**

| r | NGRDI - LAI | NGRDI - NDVI | NDVI - LAI |
|---|---|---|---|
| | 0.85 | 0.56 | 0.85 |

35

40

**Table 3: Correlation coefficients (r) between measurements of soil water content at 0.2 m and 0.5 m depth, and the surface-air temperature difference.**

| r | April 22 | June 18 | July 2 |
|---|---|---|---|
| 0.2 m | -0.56 | -0.83 | -0.74 |
| 0.5 m | -0.77 | -0.77 | -0.83 |

35

40

**Table 4: Mean values and standard deviation (in parentheses) of WDI maps along with the measured stress index (1- $\lambda E_{eddy}$ / ($R_n$-G)) and CWSI indices. CWSI$_{res}$ and CWSI$_{limit}$ computed according to (Jackson et al., 1981), Eq. (10) and Eq. (6) and (7), respectively.**

|  | April 22 | June 18 | July 2 |
| --- | --- | --- | --- |
| WDI | 0.58 (0.02) | 0.40 (0.04) | 0.41 (0.03) |
| CWSI$_{limit}$ | 0.65 (0.03) | 0.91 (0.09) | 0.85 (0.05) |
| CWSI$_{res}$ | 0.22 | 0.70 | 0.71 |
| 1- $\lambda E_{eddy}$ / ($R_n$-G) | 0.54 | 0.44 | 0.39 |

