# Peer review of "Crop water stress maps for an entire growing season from visible and thermal UAV imagery"

_Biogeosciences, 2016_

## Referee Comment (RC1) · Anonymous Referee #1 · 8 Sep 2016

General Comments: This manuscript presents an interesting case-study of the use of UAV imagery to map water stress over the course of a growing season (3 dates) in a barley field in Denmark. The authors used both RGB and thermal imagery to derive a Water Deficit Index (WDI) that relied on both land surface temperature and vegetation greenness measurements. The paper convincingly demonstrates that the addition of the vegetation greenness index to the water stress index (versus the more common approach that does not do this) is valuable for accurately mapping the spatial and temporal patterns of water stress. Overall, this paper is extremely well written, presents strong validation data, and would be a great contribution to the literature – both from a purely remote sensing perspective (UAV applications in general) and from a crop science/agricultural management perspective.

Specific Comments: I did not find many problems with this manuscript. I would like to

point out a few minor issues, though. Firstly, in Figure 4 – could the authors please explain the data gaps a bit further – why are there no gaps with the RGB imagery, but there are gaps with the thermal imagery? Secondly, It is challenging in the images presented in Figures 4-6 to clearly understand the patterns of crop ripening versus bare soil. I think it would be helpful to include some labels or annotation, as was done for some of the figures to clearly show examples of these types of locations – for example, to accompany the text descriptions on page 14?

Technical Corrections: 1) Title: Would be more accurate to state for "an" entire growing season (not seasons) since it is a single-year study. 2) Materials and Methods, 2.1 Site: Lines: 12-13 – missing word or improper word use: "while" - "The upper 0.25 m of the soil profile was homogeneous sandy loam while coarse sand with a relatively poor capacity to retain water below." 3) 2.3: Can the authors provide a citation or evidence for the following assumption? "We assumed that ripening barley had a similar green-red DN response to that of bare soil..." 4) Figure 4c – is there a better contrast stretch that can be applied to this image to display the true color? It seems highly washed out and difficult to interpret the crop pattern.

---

## Referee Comment (RC2) · Anonymous Referee #2 · 1 Nov 2016

Hoffman and others investigate early and late-season vegetation indices from a UAV platform from a barley field in Denmark. The challenge of partial canopy cover is described nicely and the manuscript is in my opinion publishable following a few minor considerations.

Where are the data from Fig. 2 from? The figures could be created in higher quality in almost all cases. Are the temperatures on the x axis representative of only a certain case? Please use standard symbols (i.e. not curly braces) in the equations.

What is most lacking in my opinion is a quantification of what, for example, "It was found that the UAV-based WDI index determines accurate crop water status" means. Was crop water status measured in the field? I noted no leaf of plant-level measurements, so such statements are not able to be validated. With respect to the conclusions, how much water should be added to stressed areas? At the moment it is a nice remote sensing-based application that struggles to be fully applicable in the field. It is publishable after the authors take more care to not over-extrapolate results to actual plant-level conditions.

———————————————————

---

## Author Comment (AC1) · 3 Nov 2016

Answer to Anonymous Referee #1 ()

Thank you for valuable and constructive feedback!

Referee comment: General Comments: This manuscript presents an interesting case-study of the use of UAV imagery to map water stress over the course of a growing season (3 dates) in a barley field in Denmark. The authors used both RGB and thermal imagery to derive a Water Deficit Index (WDI) that relied on both land surface temperature and vegetation greenness measurements. The paper convincingly demonstrates that the addition of the vegetation greenness index to the water stress index (versus the more common approach that does not do this) is valuable for accurately mapping the spatial and temporal patterns of water stress. Overall, this paper is extremely well

written, presents strong validation data, and would be a great contribution to the literature – both from a purely remote sensing perspective (UAV applications in general) and from a crop science/agricultural management perspective. Specific Comments: I did not find many problems with this manuscript. I would like to point out a few minor issues, though. Firstly, in Figure 4 – could the authors please explain the data gaps a bit further – why are there no gaps with the RGB imagery, but there are gaps with the thermal imagery?

Answer: The insufficient texture in thermal images (that lead to gaps with no data in the concatenating-process in PhotoScan), is due to a coarser resolution compared to RGB images. PhotoScan is made to be operational with RGB images, not thermal images. This will be more thoroughly explained in text related to Fig. 4.

Referee comment: Secondly, It is challenging in the images presented in Figures 4-6 to clearly understand the patterns of crop ripening versus bare soil. I think it would be helpful to include some labels or annotation, as was done for some of the figures to clearly show examples of these types of locations – for example, to accompany the text descriptions on page 14?

Answer: In a revised manuscript we will highlight locations of ripening crops and soil respectively, to better accompany points made in the text.

Referee comment: Technical Corrections: 1) Title: Would be more accurate to state for "an" entire growing season (not seasons) since it is a single-year study.

Answer: We will include an 'an' in the title, as suggested.

Referee comment: 2) Materials and Methods, 2.1 Site: Lines: 12-13 – missing word or improper word use: "while" - "The upper 0.25 m of the soil profile was homogeneous sandy loam while coarse sand with a relatively poor capacity to retain water below."

Answer: The sentence "The upper 0.25 m of the soil profile was homogeneous sandy loam while coarse sand with a relatively poor capacity to retain water below." will be

rephrased into 'The upper 0.25 m of the soil profile consisted of homogeneous sandy loam and coarse sand with a relatively poor capacity to retain water was found below'.

Referee comment: 3) 2.3: Can the authors provide a citation or evidence for the following assumption? "We assumed that ripening barley had a similar greenred DN response to that of bare soil. . ."

Answer: The sentence "We assumed that ripening barley had a similar green-red DN response to that of bare soil. . ." will be rephrased into 'Based on their color, we assumed that ripening barley and bare soil had a similar green-red DN response. . .' We hope that this rephrasing clear up uncertainties regarding the sentence.

Referee comment: 4) Figure 4c – is there a better contrast stretch that can be applied to this image to display the true color? It seems highly washed out and difficult to interpret the crop pattern.

Answer: All figures in the manuscript will be uploaded in a better quality. Figure 4c is however homogeneous and only small differences can be seen.

———————————————————

---

## Author Comment (AC2) · 3 Nov 2016

Answer to Anonymous Referee #2 ()

First, thank you for constructive and valuable feedback!

Referee comment: Hoffman and others investigate early and late-season vegetation indices from a UAV platform from a barley field in Denmark. The challenge of partial canopy cover is described nicely and the manuscript is in my opinion publishable following a few minor considerations. Where are the data from Fig. 2 from? The figures could be created in higher quality in almost all cases. Are the temperatures on the x axis representative of only a certain case? Please use standard symbols (i.e. not curly braces) in the equations.

Answer: All figures in the manuscript will be uploaded in a better quality. Fig. 2 is not based on actual data. Fig. 2 is a drawing /diagram that should illustrate the trapezoid approach; it represents a standard shape of results when plotted using this approach. This will be more thoroughly explained in the text connected to the figure in the modified version of the manuscript.

Referee comment: What is most lacking in my opinion is a quantification of what, for example, "It was found that the UAV-based WDI index determines accurate crop water status" means. Was crop water status measured in the field? I noted no leaf of plant-level measurements, so such statements are not able to be validated.

Answer: We will reword sentences like this one, where stated conclusions might extent further then what data can validate. The sentence "It was found that the UAV-based WDI index determines accurate crop water status" will be rephrased into 'It was found that the UAV-based WDI index is in agreement with measured stress values from an Eddy Covariance system.'

Referee comment: With respect to the conclusions, how much water should be added to stressed areas? At the moment it is a nice remote sensing-based application that struggles to be fully applicable in the field.

Answer: We believe that the relation between obtained WDI values and the amount of water needed, definitely is worth investigating. However, a direct link between WDI results and amount of water needed might depend on the type of vegetation and a fully and smooth applicable solution is the objective for future studies.

Referee comment: It is publishable after the authors take more care to not over-extrapolate results to actual plant-level conditions.

Answer: We will take care and rephrase concluding sentences which in the original version of the manuscript might over-extrapolate results, see for example the rephrasing in answer above, regarding 'accurate crop water status'.

---

## Author Response (AR2)

Thanks for your thoughtful comment. I have revised the manuscript and figures to ease to read for red-green colorblind colleagues. Red and green colors are not shown in same figures in the revised version.

Kind Regards Helene, on behalf of all authors.